# Transcription by the Three RNA Polymerases under the Control of the TOR Signaling Pathway in *Saccharomyces cerevisiae*

**DOI:** 10.3390/biom13040642

**Published:** 2023-04-03

**Authors:** Francisco Gutiérrez-Santiago, Francisco Navarro

**Affiliations:** 1Departamento de Biología Experimental-Genética, Universidad de Jaén, Paraje de las Lagunillas, s/n, E-23071 Jaén, Spain; 2Instituto Universitario de Investigación en Olivar y Aceites de Oliva (INUO), Universidad de Jaén, Paraje de las Lagunillas, s/n, E-23071 Jaén, Spain

**Keywords:** ribosome biogenesis, RNA polymerases, TOR pathway, transcription, *Saccharomyces cerevisiae*

## Abstract

Ribosomes are the basis for protein production, whose biogenesis is essential for cells to drive growth and proliferation. Ribosome biogenesis is highly regulated in accordance with cellular energy status and stress signals. In eukaryotic cells, response to stress signals and the production of newly-synthesized ribosomes require elements to be transcribed by the three RNA polymerases (RNA pols). Thus, cells need the tight coordination of RNA pols to adjust adequate components production for ribosome biogenesis which depends on environmental cues. This complex coordination probably occurs through a signaling pathway that links nutrient availability with transcription. Several pieces of evidence strongly support that the Target of Rapamycin (TOR) pathway, conserved among eukaryotes, influences the transcription of RNA pols through different mechanisms to ensure proper ribosome components production. This review summarizes the connection between TOR and regulatory elements for the transcription of each RNA pol in the budding yeast *Saccharomyces cerevisiae*. It also focuses on how TOR regulates transcription depending on external cues. Finally, it discusses the simultaneous coordination of the three RNA pols through common factors regulated by TOR and summarizes the most important similarities and differences between *S. cerevisiae* and mammals.

## 1. Introduction

Ribosomes are macromolecular complexes that act as ‘factories’ to carry out the neosynthesis of virtually all peptides in cells. There are about 200,000 ribosomes in an exponentially growing yeast cell, whose production rate has been estimated at ~2000 ribosomes per minute [1,2]. This scenario makes ribosome biogenesis the major source of energy use and the most important growth-related readout in the cell [1,2,3]. In fact, ~80% of total transcriptional activity in yeast and ~50% in mammalian cells is spent on producing ribosome components [4]. Considering that ribosome biogenesis is a high-demanding resource and an essential process, it must be highly regulated according to cell growth and proliferation so that when protein synthesis demand lowers, ribosome biogenesis also drastically drops [5,6,7].

The biogenesis of each yeast ribosome requires the production of the precursor of ribosomal RNA (rRNA) 35S by RNA polymerase I (RNA pol), which generates mature 18S, 5.8S and 25S rRNAs, the synthesis of 5S rRNA by RNA pol III and the expression of mRNAs (from 138 genes) by RNA pol II to generate 79 different ribosomal proteins (RPs) [5,6,7,8,9]. This process also needs the participation of several small nucleolar RNAs (snoRNAs) and more than 200 proteins known as RiBi (Ribosome Biogenesis) factors, which are involved in processing, assembling and the nuclear export of newly synthesized preribosomes, whose production also depends on RNA pol II transcription [7,8,10]. Furthermore, protein synthesis machinery also requires transfer RNAs (tRNAs) for translation, which are transcribed by RNA pol III. As cells must have an adequate proportion of the above transcripts for ribosome biogenesis and translation, the coordination of the transcription mediated by the three different RNA pols should be close [6,7,11,12]. In fact, several pieces of evidence support the existence of crosstalk among the three RNA pols [6,12,13,14,15,16,17]. However, the precise mechanisms governing this coordination are still poorly understood. It is widely accepted that the Target of Rapamycin (TOR) signaling pathway must be a key player with a central role in RNA pols regulation and ribosome biogenesis [3,5,7,8,11,18,19,20,21,22]. In this review, we focus on pre-existing knowledge and novel findings about the TOR pathway in the regulation of ribosome biogenesis through the modulation of the activity of the three RNA pols in *S. cerevisiae*. We also discuss the possible factors implicated in the simultaneous coordination of RNA pols through the TOR pathway.

## 2. The TOR Signaling Pathway: An Overview

It is essential for cells to have mechanisms that allow their metabolism and growth to be adjusted depending on external conditions. Cells respond to a wide variety of environmental cues by modulating gene expression. The TOR signaling pathway, conserved from yeast to higher eukaryotes, is one of the most important mechanisms by which cells sense nutrient availability and regulate their protein biosynthetic capacity and the ribosome production rate [3,5,7,8,19,22,23].

TOR was initially described in the budding yeast *S. cerevisiae* (reviewed in [8]). In *S. cerevisiae*, the *TOR1* gene encodes Tor1, a highly conserved Ser/Thr kinase member of the phosphatidylinositide 3-kinases (PI3K)-like family [5,8,22]. Although most eukaryotes, including mammals, only have one *TOR* gene, in *S. cerevisiae*, a paralog exists, *TOR2*, which encodes Tor2, a PI3K-like kinase with ~70% sequence homology to Tor1 [8,22,24]. Two distinct conserved complexes exist. They are termed Tor Complex 1 (TORC1) and Tor Complex 2 (TORC2) and are structurally and functionally well-differentiated [24]. However, only TORC1 is sensitive to rapamycin [24]. TORC1 is composed of Tco89, Kog1, Lst8 and either Tor1 or Tor2 [24,25,26]. It localizes mainly on the limiting membrane of the vacuole, the major nutrient reservoir organelle [27,28,29], but has also been found in the nucleus and is associated with chromatin [30,31]. TORC2 localizes at the plasma membrane and is composed of Lst8, Avo1, Avo2, Avo3, Bit61 and Tor2 [8,32]. TORC2 is also a regulator of cell growth and proliferation, although its functions do not overlap mostly with TORC1 [33]. As the TOR aspects explored in this review are mediated by TORC1, “TOR” or “TOR signaling” herein refers to the TORC1 pathway in the rest of the review.

TORC1 is sensitive to a wide variety of environmental cues, such as nutrient starvation (carbon, nitrogen, phosphate and amino acid sources), redox stress, rapamycin, among others [27,28]. The drug rapamycin is a powerful tool for studying TOR functions because it inhibits cell cycle progression and growth in yeast by binding prolyl isomerase Fpr1, which consequently forms the inactive Fpr1-TORC1 complex [24,34]. Although some questions about how TOR senses these external signals remain unanswered, when growth conditions are optimal, the TOR signaling is active, and TOR kinases regulate the activity of several proteins by direct phosphorylation [8]. In this way, TOR kinases promote cell cycle progression, nutrient import, translation and ribosome biogenesis. Nevertheless, if these environmental cues are unfavorable (i.e., nutrient starvation or rapamycin treatment), TOR kinases are inhibited, which causes the upregulation of macroautophagy, transcription repression and the downregulation of protein synthesis and ribosome biogenesis [5,19,20,21,35,36].

The signaling downstream of TOR has been classically considered on two main branches termed the Sch9 branch and the Tap42-PPases branch. Some of the important cited roles of TOR, especially in RNA pols transcription regulation, are driven by AGC family member kinase Sch9, one of the major TORC1 substrates [28]. The other important target of TOR kinases, Tap42, acts as a central regulator of PP2A and PP2A-related phosphatases [37,38,39]. In the last few years, plenty of the efforts made by many groups have focused on elucidating the precise mechanisms by which TOR regulates ribosome biogenesis. Accordingly, the direct or indirect role of TOR in the regulation of each of the three RNA pols is discussed below.

## 3. RNA pol I Transcriptional Activity Is Regulated by the TOR Pathway

rRNA production by RNA pol I represents almost 60% of overall transcription in yeast and is a limiting step for ribosome biogenesis [1]. TOR inhibition by rapamycin results in the rapid repression of RNA pol I transcription and the impairment of rRNA processing [13,21,35,40]. Nevertheless, how TOR regulates the transcription of RNA pol I is still controversial. Here, we review the RNA pol I transcription regulators linked with TOR.

### 3.1. Rrn3

TOR has been proposed to control mainly RNA pol I transcription via transcription factor Rrn3 [13,41]. This protein associates with RNA pol I subunit Rpa43 by facilitating its recruitment to promoters [42,43] Mechanistically, rapamycin causes a decrease in Rrn3-RNA pol I association and, thus, the release of RNA pol I from rDNA promoters [13,41]. In addition, Rrn3-RNA pol I binding depends on both Rrn3 and RNA pol I phosphorylation status [44,45]. Both *RRN3* and *RPA43* are essential genes but can be deleted in a strain that overexpresses fusion protein Rrn3-Rpa43 (termed CARA protein) [13]. In CARA cells, 35S rRNA synthesis is not severely affected after rapamycin treatment, which suggests that Rrn3-RNA pol I interaction is the major target of TOR to regulate rDNA transcription [13]. Another suggestion is that two other RNA pol I subunits (Rpa49 and Rpa34) participate not only in the recruitment of Rrn3 to rDNA promoters but also in its subsequent release from the RNA pol I complex during elongation [46]. It can be speculated that TOR may target these subunits to regulate Rrn3-RNA pol I association and activity.

RNA pol I recruitment to rDNA promoters is primarily mediated by Sch9 (and, to a lesser extent, by Tap42), one of the TORC1 major targets, a process that is inhibited by rapamycin [40]. However, the precise mechanisms mediating this process remain unknown. Notably, Rrn3-RNA pol I dissociation upon TOR inhibition seems to be independent of Sch9 activity [40]. In addition, TOR inactivation leads to the proteasome-dependent degradation of Rrn3, which correlates with the impaired formation of Rrn3-RNA pol I complexes and rRNA neosynthesis [47]. By using nondegradable Rrn3 expressing strains, the authors have demonstrated that Rrn3 degradation does not fully explain the marked downregulation of rRNA production upon TOR inhibition [47]. This observation suggests that additional steps may be required for rDNA transcription downregulation after TOR inhibition. Although, other authors propose that rDNA transcription downregulation after TOR inhibition is mostly an indirect consequence of reduced RPs production [48]. Nevertheless, other potential targets downstream of TOR and a more direct role of TORC1 in RNA pol I transcription have been subsequently demonstrated.

### 3.2. Hmo1

The high-mobility group B (HMG) family member Hmo1 was identified as a factor involved in rDNA transcription by genetic interactions between *HMO1* and *RPA49*, which encodes Rpa49 [49]. Hmo1 localizes closely to the nucleolus [49] and associates at many locations throughout the rDNA genes [50]. Hmo1 dissociates from rDNA promoters upon rapamycin treatment [51], and *HMO1* expression is regulated in a TOR-dependent manner [52], with the participation of Tor1, which binds directly to the *HMO1* gene [53]. Taken together, these data suggest that TOR might modulate Hmo1 levels to control rRNA production.

### 3.3. Ccr4-Not

The Ccr4-Not complex is a well-known element involved in several RNA pol II-dependent gene expression steps (reviewed in [54]). Ccr4-Not has been related as a TOR pathway-crosstalk element, an activator and a downstream factor (reviewed in [55]). Genetic approaches have identified some Ccr4-Not mutants as being rapamycin-sensitive, which indicates the relation between this complex and the TOR pathway [56,57]. In fact, Ccr4-Not may act upstream of TOR through a mechanism that involves V-ATPase [58]. Ccr4-Not associates with both RNA pol I and rDNA, which indicates a role in RNA pol I regulation [59]. Notably, Ccr4-Not disruption counteracts Rrn3 dissociation from the RNA pol I enzyme upon TOR inactivation by preventing RNA pol I repression [59]. These data suggest a role for the Ccr4-Not complex, that of bridging TOR with RNA pol I transcription [59].

### 3.4. Paf1 Complex

The RNA polymerase-associated factor 1 complex (Paf1C) is a well-known transcription factor involved in RNA pol II transcription [60]. A role for Paf1C during RNA pol I transcription has been demonstrated: Paf1C associates with rDNA and promotes transcription by RNA pol I, probably by enhancing the elongation rate [61,62]. In vitro approaches also confirm this conclusion [62]. Interestingly, overall transcription is not affected in *paf1Δ* cells after histidine starvation [62], and rapamycin treatment only causes partial RNA pol I transcription reduction and does not lead to RNA pol I dissociation from rDNA in *paf1Δ* cells compared to the wild-type [62]. These findings suggest that Paf1 is required for RNA pol I control in response to stress signals. Collectively, all these results indicate Paf1C as a factor that positively modulates RNA pol I transcription under optimal growth conditions but is also required for RNA pol I repression after TOR inhibition [62].

### 3.5. TOR Mediates Epigenetic and Chromatinic Changes to Control RNA pol I Activity

It has been estimated that the number of chromosomal rDNA copies in *S. cerevisiae* is ~150, half of which are transcriptionally active in exponentially growing cells [63]. Growth conditions influence the number of active rDNA copies [64,65], and cell volume modulates the rDNA repeat copy number and the rRNA synthesis rate [66]. Although some authors propose that TOR does not influence the number of active rDNA genes, it is tempting to speculate that TOR might play a role in the regulation of these processes [41]. TOR positively influences rDNA amplification depending on nutrient availability [67]. Furthermore, TOR inhibition leads to the increased recruitment of histone deacetylase Rpd3 (an RPD3L histone deacetylase complex subunit) to rDNA regions by decreasing the acetylation of histone H4 K5/K12 residues and, thus, causing rDNA chromatin condensation and smaller nucleolar size, which affect RNA pol I transcription [68].

Several histone H3 modifications are positively regulated by TOR [69]. Of them, histone H3 lysine 56 acetylation (H3K56ac) is a chromatin modification that influences cell growth through rRNA synthesis [69]. TOR inhibition decreases H3K56ac by causing the lesser association of Hmo1 with rDNA and, thus, impairs RNA pol I transcription [69]. The H3K56ac modification is mediated by histone chaperone Asf1 and acetyltransferase Rtt109 [70,71]. As both *asf1Δ* and *rtt109Δ* mutants show hypersensitivity to rapamycin [69], TOR could control histone modifications through a pathway that involves Asf1 and Rtt109 [69].

Tor1 kinase shuttles between the nucleus and the cytoplasm [30]. Nutrient deprivation or rapamycin treatment leads Tor1 to leave the nucleus, which results in decreased rRNA synthesis [30]. Tor1 and Kog1 subunits associate directly with 35S gene promoters, which suggests that TORC1 could exert its activity in chromatin, possibly by modulating the condensation of rDNA repeats [30]. We hypothesize that this process could occur through the phosphorylation of transcriptional machinery, which could involve RNA pol I. In fact, overall phosphorylation of pol I is important for transcription initiation [45], and 115 different phosphosites in all 14 RNA pol I subunits have been identified, despite the direct participation of TOR not yet having been demonstrated [72].

Taken together, these data support that TOR acts at different levels to control RNA pol I activity and, hence, ribosome biogenesis. The roles of the TOR pathway in RNA pol I transcription are summarized in Figure 1.

## 4. Expression of RPs and RiBi Genes by RNA pol II Is Regulated by TOR

About 50% of all RNA pol II-mediated transcription events are devoted to the production of RPs [1]. The TOR pathway influences RNA pol II activity to modulate ribosome biogenesis. TOR influences the expression of RNA pol II-transcribed genes by controlling the nuclear/cytoplasmic localization of transcription factors through its phosphorylation regulation [5,7,18,36]. Here we discuss those directly related to RPs and RiBi transcription, some of which act in response to environmental conditions.

### 4.1. Rap1, Fhl1 and Ifhl1/Crf1

Transcription factor Rap1 is a key player for RPs and RiBi gene expression. It binds most RP gene promoters and, hence, accounts for ~50% of total RNA pol II-mediated transcription [1]. Rap1 recruits forkhead-like transcription factor Fhl1, and both are constitutively bound to RP promoters [17,73,74,75]. Fhl1 is, in turn, regulated by Ifh1 (co-activator) and Crf1 (corepressor) proteins [17,74,75]. Under favorable growth conditions, TORC1 maintains corepressor Crf1 in the cytoplasm (via Yak1 kinase) to, thus, prevent its interaction with Fhl1 and allow the recruitment of phosphorylated Ifh1 to the Rap1-Fhl1 complex by driving RPs gene expression [17,73,74,75]. Upon nutrient deprivation, Crf1 is phosphorylated by Yak1, shuttles into the nucleus and competes with Ifh1 to bind Fhl1, which brings about the downregulation of RPs gene expression [75]. This correlates with Ifh1 being released from promoters after rapamycin treatment but has no effect on Rap1 and Fhl1 promoter occupancy [17,74,75]. The phosphorylation of both Ifh1 and Crf1 by Casein Kinase 2 (CK2), a downstream effector of TORC1 [76], has been reported as being important for the binding to Fhl1 at RP gene promoters [77]. Furthermore, acetylation and phosphorylation of Ifh1 are inhibited by rapamycin treatment [78].

### 4.2. Sfp1

Split zinc (Zn)-finger protein Sfp1 participates in RP and RiBi gene expression in a TOR-dependent manner [79,80,81]. Its contribution to RP gene transcription is gene-specific because its nuclear depletion leads to the marked downregulation of the expression of some RP genes, with a milder impact on others [82]. In addition, Sfp1 associates with RiBi promoters, and the anchoring of Sfp1 causes the downregulation of RiBi gene expression [83].

Sfp1 rapidly translocates from the nucleus to the cytoplasm following the stress conditions that lead to the downregulation of RP and RiBi gene expression [79,80,81]. Sfp1 also interacts and is directly phosphorylated by TORC1 under optimal growth conditions, which allows Sfp1 nuclear localization and the association with RP gene promoters [81]. Interestingly, Sfp1 negatively regulates TORC1 activity toward the Sch9 effector, which suggests the existence of a feedback mechanism that links RP and RiBi gene expression with TOR signaling and Sch9 activity [81].

### 4.3. Abf1

General transcription factor Abf1 has a well-known role as an activator of RP and RiBi gene expression under optimal nutrient conditions by binding a small group of RP and ~50% of RiBi gene promoters [84,85]. However, its occupancy largely increases upon TORC1 inactivation [84,85]. Furthermore, rapamycin treatment changes neither the expression levels nor the subcellular localization of Abf1, which suggests that increased Abf1 occupancy in promoter regions results from a change in its DNA-binding properties, possibly due to a modification of its phosphorylation status via TORC1 [86,87]. In fact, its phosphorylation varies in accordance with nutrient availability [88]. As Abf1 positively regulates RP and RiBi gene expression under nutrient-rich conditions, the question of why TOR inhibition should cause increased Abf1 occupancy is still unanswered.

### 4.4. TOR Regulates RP Gene Expression through Hmo1

Hmo1 binds to the rDNA gene body and plays a well-known role in RNA pol I transcription (see Section 3.2). Interestingly, Hmo1 also binds most RP gene promoters [50], although the effect of Hmo1 on RP gene expression remains controversial. One report shows *HMO1* deletion does not lead to the overall downregulation of RP genes compared to wild-type cells [50]. Shortly afterward, however, another report found a mild but significant decrease in RP gene expression after Hmo1 depletion [51]. Moreover, the absence of Hmo1 imbalances the expression of some RP genes, which suggests that it could coordinate the expression of these genes to ensure an adequate proportion of transcripts for ribosome biogenesis [51].

As with RNA pol I, Hmo1 dissociates from RP gene promoters after rapamycin treatment, which correlates with the downregulation of RP genes [51]. Notably, rapamycin causes the upregulation of RP gene expression in *hmo1Δ* cells compared to the wild-type, which indicates that Hmo1 is also required for the repression of RP genes in a TOR-dependent manner [51].

### 4.5. TORC1 Controls RNA pol II Activity through the Sch9 Downstream Effector

The downstream kinase effector of TOR, Sch9, positively regulates RiBi and RP gene expression in a TORC1-mediated manner [28,79]. Sch9 could phosphorylate some transcriptional elements in order to regulate RiBi and RP genes, such as Rap1 [79]. Indeed, Sch9 phosphorylates RNA pol II transcriptional repressors Dot6, Tod6 and Stb3 to thus, favor transcription [89,90]. In line with this, Dot6, Tod6 and Stb3 are dephosphorylated after TOR inhibition. This allows their binding to the promoter regions of RiBi and RP genes and the recruitment of the RPD3L histone deacetylase complex, which leads to the repression of the Ribi/RP gene promoters (see Section 4.6. for more information) [89,91].

### 4.6. TORC1 Regulates RNA pol II Transcription by Chromatin Remodeling and Modification

As with RNA pol I, TOR plays a role in chromatin remodeling during RNA pol II-mediated transcription. The RSC (Remodeling the Structure of Chromatin) complex is an essential ubiquitous and ATP-dependent chromatin remodeler that performs nucleosome repositioning to activate or repress transcription [92]. After TOR inhibition, RSC subunit Rsc9 shows an altered localization profile throughout the genome, including a reduction in the occupancy of the genes involved in ribosome biogenesis [93]. This suggests that RSC could act as a downstream effector of TOR to mediate stress signals with chromatin remodeling in several genes, including RP genes. Moreover, the INO80 chromatin remodeling complex has been proposed as a downstream effector of TOR for ribosome biogenesis [94]. The deletion of *INO80*, which renders *ino80Δ* cells resistant to rapamycin, partially mimics the ribosome biogenesis genes’ transcriptional profile observed in rapamycin-treated cells [94].

As previously indicated, TOR seems to play a role in histone modification. TOR regulates the recruitment of Esa1 (the catalytic subunit of the NuA4 histone acetylase complex) to RP gene promoters. This allows histone H4 acetylation, which activates the transcription of these genes [95]. Accordingly, rapamycin treatment causes Esa1 release from RP gene promoters and transcription repression [95]. Conversely, the mutants of the RPD3L histone deacetylase complex exhibit impaired RP gene repression after TOR inhibition [95]. As deacetylation is needed for gene repression, this result suggests that TOR could regulate RPD3L activity in RP genes. Furthermore, Rpd3 has been proposed to be recruited to RP promoters in a TORC1-dependent manner [91], although it could be constitutively associated [95].

The roles of TOR in RNA pol II regulation for ribosome biogenesis are schematized in Figure 2.

## 5. The TOR Signaling Pathway Regulates RNA pol III Transcription

RNA pol III is specialized in the high-rate production of short RNAs through several rounds of reinitiation over the DNA template. It synthesizes rRNA 5S, which is needed for ribosome biogenesis and all the tRNAs required for translation. To synthesize these transcripts, the RNA pol III holoenzyme needs the participation of two different basal transcription factors, termed TFIIIB (composed of TATA-binding protein TBP, Brf1 and Bdp1), TFIIIC (composed of Tfc1, Tfc3, Tfc4, Tfc6, Tfc7 and Tfc8), among other proteins (reviewed in [96]). In 5S rRNA synthesis, RNA pol III also requires the participation of transcription factor TFIIIA (also known as Pzf1 or TFC2) [96].

As with RNA pols I and II, rapamycin treatment disturbs RNA pol III transcription and causes the downregulation of RNA pol III-dependent genes [35,89]. How does TOR influence RNA pol III transcription? Most RNA pol III transcriptional control relies on the Maf1 repressor, although novel findings have shown that other RNA pol III machinery elements are also involved.

### 5.1. Maf1: Phosphoregulation and Activity

Maf1 is the master regulator of RNA pol III-mediated transcription, conserved from yeast to humans (reviewed in [97]). Indeed, *maf1Δ* cells are defective in the downregulation of RNA pol III transcription under different growth conditions, including TOR inhibition [98]. Co-immunoprecipitation assays have confirmed that Maf1 specifically interacts with RNA pol III subunits [99,100,101,102] and also with TFIIIB component Brf1 [99]. Notably, the Maf1-RNA pol III interaction largely increases upon TOR pathway inhibition [101,102].

Maf1 activity and cellular localization are regulated by phosphorylation (reviewed in [97]). Under favorable growth conditions, the protein is phosphorylated and predominantly located in the cytoplasm, but a small portion remains nuclear ([103,104] and our unpublished data). Under unfavorable growth conditions, Maf1 is largely dephosphorylated and accumulates in the nucleus. Phosphorylated Maf1 is exported from the nucleus to the cytoplasm by the action of exportin Msn5 [103]. Notably, in *msn5Δ* cells, Maf1 is constitutively located in the nucleus but does not cause the downregulation of RNA pol III transcription [103]. This indicates that Maf1 dephosphorylation is the critical mechanism for RNA pol repression and not its nuclear localization. Indeed the dephosphorylated state of Maf1 correlates with both an increased Maf1-RNA pol III interaction and the dissociation of RNA pol III from the DNA template [101,102,105]. Maf1 binding leads to a rearrangement of the C82-C34-C31 heterotrimer, which renders Maf1-RNA pol III complexes unable to interact with TFIIIB and be recruited to promoters [106]. The kinases and phosphatases involved in Maf1 regulation are discussed below.

#### 5.1.1. Sch9

Many studies have pointed out that Sch9 is the major kinase of Maf1 under optimal growth conditions ([40,107,108] and our unpublished data). Sch9 mainly localizes in the cytoplasm and is associated with the vacuolar membrane [28,109,110]. Sch9 activity is required for high RNA pol III gene transcription levels, and the deletion of *MAF1* in *sch9Δ* cells restores transcriptional defects [107]. This suggests that Sch9 could facilitate RNA pol III transcription by phosphorylating Maf1. Furthermore, *SCH9* deletion renders dephosphorylated Maf1 and causes its nuclear localization [108]. Sch9 interacts with Maf1 and phosphorylates the seven serines identified as phosphosites of Maf1 in vitro [40,108]. Yet does Sch9 kinase activity act on Maf1 via TOR? The expression of a constitutively active allele of *SCH9* (termed *SCH9^DE^*), which uncouples Sch9 activity from the upstream regulation by TORC1, blocks the repression of RNA pol III transcription [40,108] and prevents Maf1 from interacting with RNA pol III after rapamycin treatment [40]. These data indicate that RNA pol III repression, carried out by Maf1, requires the signaling from TOR to Sch9. In fact, TORC1 phosphorylates Sch9 in a nutrient-dependent manner [28]. Interestingly, RNA pol III activity is still sensitive to rapamycin treatment in the cells expressing Maf1^7E^, a phosphomimetic version of native Maf1. This led the authors to suggest that Sch9 could target additional factors to promote RNA pol III transcription [40]. Another possibility that cannot be ruled out is that Maf1 could have additional phosphosites that have not yet been identified. Whether TORC1 could regulate Maf1 phosphorylation in not only Sch9-dependent but also in an Sch9-independent manner is discussed below.

#### 5.1.2. PKA: Crosstalk with Sch9 and TOR Pathway

Six of the seven phosphosites of Maf1 correspond to consensus Sch9/PKA sites [111], which indicates that Maf1 can be targeted by Sch9 and cyclic AMP-dependent Protein Kinase A (PKA). In fact, Maf1 has primarily been identified as a substrate of PKA by a proteomic approach, and PKA phosphorylates Maf1 in vitro [112]. PKA is an important glucose-sensitive regulator of cell growth, metabolism, stress response and ribosome biogenesis in *S. cerevisiae* [113]. PKA acts in the Ras/cAMP (also known as Ras/PKA) signaling pathway that is conserved throughout eukaryotes [114]. The overexpression of either Tpk1 or Tpk2 catalytic subunits causes the hyperphosphorylation of Maf1 [107], and marked PKA activity blocks RNA pol III repression [104] consistently with a negative role of PKA activity in the Maf1 function.

Some questions arise about the impact of Sch9 and PKA on Maf1 phosphoregulation. Is the activity of both Sch9 and PKA redundant, or does PKA operate in parallel in an independent pathway to modulate the Maf1 function? We can hypothesize that those pathways cooperate because not only have several genetic interactions been reported between elements of them both (reviewed in [115]), but crosstalk between both pathways has also been proposed, likely through convergence on shared substrates [115,116]. Furthermore, although both pathways can be considered to operate in parallel to promote growth in response to different nutritional signals [90,117,118], in some cases, the inhibition of one pathway can result in the compensation or hyperactivation of the other [119].

Some observations suggest that Sch9 can phosphorylate the PKA consensus sites of Maf1, even when PKA activity is lacking, but this redundant effect is not bidirectional [40,107]. These data hint at Sch9 being the major kinase of Maf1. It has also been suggested that Sch9 operates upstream of PKA because *SCH9* deletion leads to Bcy1 (the regulatory PKA subunit) hyperphosphorylation, which attenuates PKA activity [120]. Finally, rapamycin causes Tpk1 catalytic subunit nuclear accumulation in an Sch9-dependent manner [120].

The overall data suggest that TORC1 acts on Sch9 to regulate Maf1 phosphorylation under favorable growth conditions, while the Ras/PKA pathway probably provides an additional mechanism to control Maf1 activity through sensing glucose availability. Whether PKA has a stronger impact on Maf1 phosphoregulation under other growth conditions cannot be ruled out.

#### 5.1.3. Casein Kinase 2

The CK2 enzyme is an essential serine/threonine kinase involved in numerous important cellular processes [121]. CK2 activity is required for a robust RNA pol III transcription [122], and it has been reported as a downstream target of TOR [76].

Cka2-Maf1 interaction has been reported in a carbon source-dependent manner, and CK2 phosphorylates Maf1 both in vitro and in vivo [105]. Notably, CK2 is required to activate RNA pol III activity when transferring cells from repressive to favorable growth conditions because it phosphorylates Maf1 by promoting its dissociation from the RNA pol III enzyme and its export from the nucleus to the cytoplasm [105]. However, the involvement of CK2 in this process remains controversial because a mutant lacking all the CK2 phosphosites in Maf1 has no effect on RNA pol III reactivation upon the transfer from glycerol- to glucose-containing medium [123]. Nonetheless, these data do not exclude the possibility of CK2 targeting other RNA pol III machinery elements during transcription. Indeed CK2 phosphorylates TBP to promote RNA pol III transcription (see Section 6 for further information) [122,124].

#### 5.1.4. Protein Phosphatase 4

TOR inhibition causes the rapid dephosphorylation of Maf1, which leads to Maf1-RNA pol III interaction and transcription repression [101,102]. In line with this, the Protein Phosphatase 4 (PP4) complex has been reported as the major phosphatase of Maf1 [125].

PP4 phosphatase is an evolutionary conserved Ser/Thr phosphatase that is involved in several essential processes in eukaryotic cells [126]. In *S. cerevisiae*, the PP4 core complex is composed of three subunits: Pph3, the catalytic subunit; Psy2, the regulatory one; Psy4, the regulatory/scaffold subunit [127]. Deletion of *PPH3* or *PSY2* genes leads to the impairment of both Maf1 dephosphorylation and RNA pol III repression after TOR inhibition, which falls in line with the role of PP4 in Maf1 regulation ([125] and our unpublished data). Maf1 interacts with Pph3 in both log-phase and stressed cells ([125] and our unpublished data). This indicates that a fraction of total Maf1 seems to be constitutively bound to PP4. Notably, the lack of the Psy4 subunit does not affect either Maf1 dephosphorylation or the Pph3-Psy2 interaction, which indicates a minor role of Psy4 in these processes and suggests that PP4 could exist in several subcomplexes in vivo [125]. As PP4 belongs to the phosphoprotein phosphatases family (PPPs), one hypothesis yet to be tested is that TOR kinases or the Sch9 effector could control PP4 catalytic activity through the direct phosphorylation of some of its subunits. Although very little is known about PP4 biogenesis, regulation and mechanism of activation, we can hypothesize that Maf1 activation may occur via conformational and/or posttranslational modifications of PP4 to improve its catalytic activity.

### 5.2. TORC1 Directly Controls RNA pol III Transcription

TOR signaling controls RNA pol III gene expression indirectly through some effectors, mainly Sch9. Nevertheless, TORC1 can also regulate RNA pol III transcription in an Sch9-independent manner [107]. A direct role of TORC1 in RNA pol III transcription has also been reported [31]. TORC1 (Tor1 and Kog1 subunits) binds to 5S rDNA promoters in a nutrient-dependent manner [30,31], and although it does not associate with any tested tDNA [31], it seems to indirectly control tRNAs neosynthesis. Furthermore, unfavorable growth conditions cause the dissociation of Tor1 and Kog1 from chromatin [31]. 

TORC1 associates with Maf1 in vivo and phosphorylates Maf1 in vitro [31]. A model for TORC1 regulation over RNA pol III gene expression has been proposed: under favorable growth conditions, TORC1 associates with 5S rDNA promoters and directly phosphorylates Maf1, which prevents its nucleolar accumulation, and leads to Maf1 export from the nucleus to the cytosol [31].

### 5.3. Other RNA pol III Machinery Elements as Targets of TOR

The cells expressing the nonphosphorylable Maf1^7A^ version of Maf1 do not exhibit constitutively repressed RNA pol III and are still strongly downregulated upon rapamycin treatment [40]. These interesting observations indicate that another mechanism may influence RNA pol III regulation. This poses the question of whether other RNA pol III machinery elements are involved in TOR-mediated transcriptional control.

#### 5.3.1. Rpc53

Rpc53 (an RNA pol III specific subunit) phosphorylation is TOR-dependent [128]. Two kinases downstream of TOR, Kns1 and Mck1 are responsible for Rpc53 phosphorylation under unfavorable growth conditions [128]. The deletion of either *MCK1* or *KNS1* avoids Rpc53 phosphorylation after rapamycin treatment, which correlates with the impairment of RNA pol III repression [128]. Kns1 is the sole member of the LAMMER/Clk kinase family in yeast [129]. Mck1 is one of the four Glycogen Synthase Kinase-3 (GSK-3) homologs in *S. cerevisiae* that participate in several important cellular processes [130].

Interestingly, adding rapamycin to *maf1Δ* cells causes similar RNA pol III repression to the triple mutant *maf1Δ kns1Δ mck1Δ*, which indicates that Kns1 and Mck1 kinases do not seem to regulate RNA pol III independently of Maf1 [128]. Therefore, an integrative model has been proposed in which TOR inhibition causes the upregulation of Kns1 expression, which leads not only to its autophosphorylation and nuclear accumulation but also to Rpc53 phosphorylation, which later favors the subsequent Rpc53 hyperphosphorylation by Mck1. Consequently, hyperphosphorylated Rpc53 can disrupt the recycling of RNA pol III to the promoter by allowing the subsequent Maf1 interaction and repression [128].

#### 5.3.2. Bdp1

Bdp1 is an essential subunit of the TFIIIB complex and is required for RNA pol III recruitment to promoters [131]. Notably, Bdp1 is the only subunit of TFIIIB that has no counterpart in RNA pol I or II transcription machinery (reviewed in [96]).

Bdp1 phosphorylation (and thus, its activity) seems to be TOR-dependent [132]. Indeed, in vitro and in vivo experiments show that Bdp1 is targeted by several kinases, including Sch9 [132]. This is consistent with the notion that Sch9 may target other substrates that differ from Maf1 to enable RNA pol III transcription [40]. Although the biological significance of Bdp1 phosphorylation is not fully defined, the data suggest that it contributes to transcription by facilitating RNA pol III recycling and antagonizing the repression by Maf1 [132].

The roles of TOR in the regulation of RNA pol III are schematized in Figure 3.

## 6. Crosstalk of the Three RNA pols Could Be Mediated by TOR

Although crosstalk among the three RNA pols has been proposed, whether RNA pols are temporally and spatially regulated through the TOR pathway remains unclear. Sch9 indirectly influences not only the transcription of RNA pol III by phosphorylating Maf1 ([40,107,108] and our unpublished data) and Bdp1 [132] but also the expression of the RP and RiBi genes by RNA pol II by targeting repressors Dot6, Tod6 and Stb3 [89]. However, although RNA pol I recruitment to promoters and transcription seems to be influenced by Sch9, the details of this mechanism remain unknown [40].

Transcription coordination could theoretically be achieved via the regulation of some shared subunits of the three RNA pols, which could occur through posttranslational modifications. Indeed, phosphoproteomic approaches have identified 20 different phosphosites in the shared subunits of the three RNA pols (reviewed in [72]). However, both the biological significance of these putative phosphorylations and the relation with TOR are unknown. In line with this, Rpb5, a common subunit of the three RNA pols, interacts with unconventional prefoldin Bud27 [133,134]. Bud27, the only known factor to participate in the cytoplasmic assembly of the three RNA pols in an Rpb5-dependent manner [133,135], along with its human orthologue URI, are TOR pathway elements [134,136]. Moreover, human URI is phosphorylated depending on mTOR signaling [134], while Bud27 contains some putative phosphosites. We have recently reported that a lack of Bud27 impairs the transcription of the three RNA pols, which may occur, at least partially, due to TOR pathway alteration [12]. This interesting finding points out that Bud27 is a factor that might simultaneously coordinate the transcription of the three RNA pols, probably by association with Rpb5.

Another possibility is that TOR could control the transcription of the three RNA pols via a common transcription factor. TBP is a universal transcription factor that mediates the transcription of the three RNA pols [137]. A phosphoproteomic analysis has shown potential phosphosites in TPB [132]. TBP is known to be phosphorylated by CK2 [122,124], whose activity is, in turn, regulated by TOR [76]. Yet, the biological impact of these phosphorylations has only been reported in the RNA pol III context, and no further research works into other RNA pols are available. In line with the notion of TOR regulating general transcription factors’ activity, a recent report indicates that the demethylase Rph1 histone negatively modulates RNA pols I and II transcription under nutrient stress conditions [138]. *RPH1* and *TOR1* genetically interact, and *rph1Δ* cells exhibit rapamycin resistance and defects in RNA pol I and II repression, which is consistent with a negative role of Rph1 under TOR-inhibiting conditions [138]. Under optimal growth conditions, Rph1 associates with 35S rDNA, RP and RiBi, but not with RNA pol III-mediated genes. Conversely, TOR pathway inactivation causes the phosphorylation of Rph1 and its dissociation from rDNA and the RPs genes locus by attenuating transcription [138]. Thus, TOR could regulate Rph1, which, under repressive conditions, could negatively modulate RNA pols I and II to ensure cell survival [138]. Yet whether Rph1 also acts on RNA pol III remains to be elucidated.

Cells can adjust the transcription of rDNA in response to the downregulation of RP genes [13]. In CARA cells, unaffected RNA pol I activity can alleviate the TOR-dependent regulation of RP genes after rapamycin treatment, which suggests crosstalk between RNA pol I and II [13]. Interestingly, the deletion of *HMO1* in CARA cells largely alleviates the upregulation of most RP genes upon rapamycin treatment [51]. This finding, combined with the data showing Hmo1 binding both the rDNA genes and also most of the RP gene promoters, suggests a role of Hmo1 in coupling RNA pol I and II machinery for ribosome biogenesis [51]. However, a role for Hmo1 in RNA pol III transcription has not yet been demonstrated.

Another intriguing hypothesis is that TOR could simultaneously regulate the activity of the three RNA pols for optimal ribosome biogenesis through chromatin remodeling. As mentioned in Section 4.6, TOR inhibition leads to the altered localization of Rsc9 throughout the genome [93]. This alteration also occurs in those genes related to the ribosome biogenesis process [93]. Therefore, RSC could act downstream of TOR to regulate gene expression, at least for RNA pol II-mediated transcription. In line with this, a role for RSC in RNA pol III-mediated transcription has been proposed [139,140], and the RSC-RNA pol I interaction has been reported in vivo [140], but whether this association can influence rDNA transcription remains unknown.

## 7. Comparison of RNA pols Regulation by TOR between *S. cerevisiae* and Mammals

The TOR signaling pathway is highly conserved throughout eukaryotes. Unlike yeast, all other eukaryotes have only one Tor protein (termed mTOR in mammals), which forms two TOR complexes: mTORC1 and mTORC2 [141]. As in yeast, only mTORC1 is sensitive to rapamycin [141] and is also a key regulator of transcription for ribosome biogenesis [5,7,142,143,144,145].

rDNA transcription regulation by mTORC1 is not fully understood, although the global process seems conserved. Like yeast, mTORC1 inhibition leads to a rapid decrease in rRNA production [146,147], and TIF-1A and UBF (Upstream Binding Factor 1), the mammalian orthologs of Rrn3 and Hmo1, respectively, are the main regulators of RNA pol I activity [5,7,143,144,145]. Rapamycin causes changes in TIF-1A phosphorylation, which leads to its cytoplasmic accumulation by causing the impairment of the TIF-1A-RNA pol I interaction and, thus, rDNA transcription [148]. Moreover, mTORC1 regulates the phosphorylation (and, thus, activity) of UBF by promoting the rDNA transcription [149]. Notably, the phosphorylation of UBF depends on S6 kinase (S6K) activity [149], the mammalian ortholog of yeast Sch9 and a substrate of mTORC1 [150]. Finally, and as with yeast, mTORC1 also associates with rDNA promoter regions [151], although the biological impact of this interaction remains unknown.

In *S. cerevisiae*, RPs, and RiBi gene expression are regulated by TOR via different mechanisms that directly or indirectly influence RNA pol II activity. In contrast, mammalian mTORC1 regulates RPs production mainly at the translational level [142]. mTORC1 promotes the translation of mRNA by encoding all RPs and also some translation elongation factors (reviewed in [145]). These pieces of evidence suggest that mammalian cells could have evolved to tightly regulate RP production, preferably at the posttranscriptional level instead of transcriptionally. Intriguingly, these mechanisms seem to differ for RiBi production because mTORC1 regulation is exerted transcriptionally via S6K [152].

mTORC1 inhibition causes RNA pol III transcription attenuation [153]. This mechanism is, at least partially, evolutionary conserved. Like yeast, RNA pol III regulation in mammalian cells relies on the MAF1 repressor. mTORC1 inhibition leads to MAF1 dephosphorylation [154,155,156,157]. Strikingly, and unlike yeast Maf1, mammalian MAF1 does not contain consensus sites for the Sch9 homolog S6K, which suggests that its phosphoregulation may be exerted by another kinase, which could be mTORC1 because their direct association has been reported [155,158]. mTORC1 also associates with RNA pol III genes. Yet, unlike yeast, it also appears enriched in tDNA genes [151,156,158]. Furthermore, a model has been proposed in which TFIIIC recruits mTORC1 to DNA regions that, in turn, phosphorylates and excludes MAF1 from chromatin [158]. Note that no interaction between yeast TORC1 and TFIIIC has yet been reported.

The transcriptional coordination of the three RNA pols by mTORC1 has also been proposed in mammals, but the mechanisms governing this process are still poorly understood [5]. One of the elements that allow this coordination could be MAF1 because it has been reported as not only an RNA pol III regulator but also as an RNA pol I and II [159]. Other putative candidates for such coordination are ubiquitous factors c-Myc, p53 or TBP because they participate in the transcription of the three RNA pols [7,11,160,161]. Finally, we can also propose URI, the human ortholog of yeast Bud27, as a coordinator. URI has been reported as a factor that coordinates the mTOR pathway with gene expression [134], as with yeast [12], and possibly by its interaction with RNA polymerase common subunit RPB5 [133].

## 8. Conclusions

Ribosome biogenesis regulation at the transcriptional level has become an interesting and exciting research field. There is growing evidence that cells require the tight regulation of ribosome production to properly respond to fluctuating environmental cues, and signaling pathways play a critical role in this process. This review overviews the transcriptional regulation of the three RNA pols by the highly conserved TOR signaling pathway. To date, some transcription factors have been identified as effectors of TOR to modulate the activity of each RNA pol. Understanding the mechanisms by which TOR regulates these factors has been an important step to further comprehend RNA pol transcription regulation, ribosome biogenesis and cell growth. Interestingly, recent advances in this field have led to the hypothesis about the factors that could simultaneously coordinate RNA pols under the control of TOR. These mechanisms could include modifications in the common subunits of RNA pols, the factors that target these subunits, general transcription factors, global epigenetic modifications and chromatin remodeling. Further research is needed to examine the mechanisms that govern these processes. Furthermore, crosstalk between RNA pols must exist, or even between RNA processing and transcription. Future exploratory perspectives may include the existence of feedback loops to finely tune and regulate RNA homeostasis and environmental fluctuations.

## Figures and Tables

**Figure 1 biomolecules-13-00642-f001:**
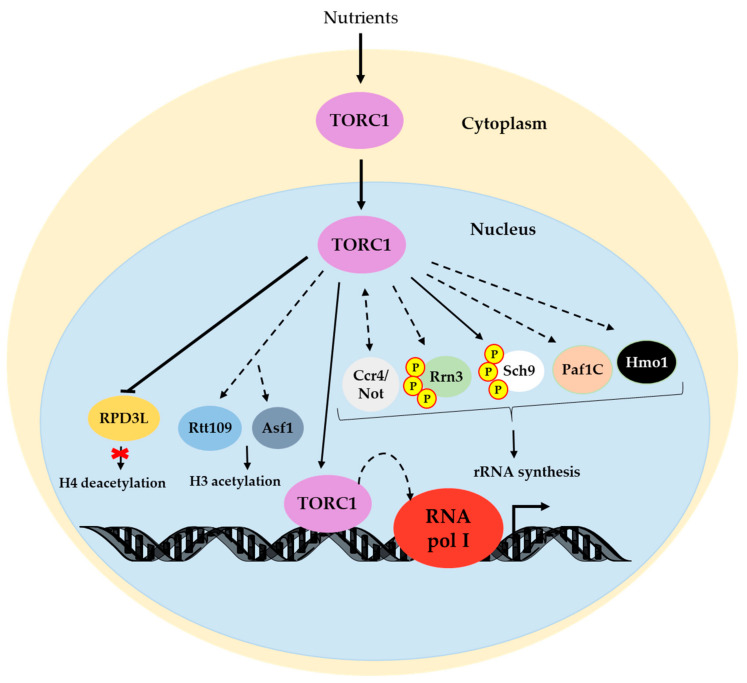
Regulatory network from TORC1 to RNA pol I transcription. Under nutrient-rich conditions, TORC1 localizes in the nucleus and also associates with chromatin to possibly favor transcription [30]. TOR regulates histone modification by regulating Asf1 and Rtt109 positively and RPD3L negatively [68,69]. Nuclear TOR also promotes the Rrn3-RNA pol I interaction [13,41] and indirectly influences RNA pol I transcription through Sch9 [40], Paf1C [62], Hmo1 [51] and Ccr4/Not [59]. Note that bidirectional arrows are represented in the last case due to crosstalk between that element and TOR [55]. Solid lines denote direct positive regulation. Dotted lines depict indirect positive regulation. Bars refer to negative regulation. Details for each mechanism are provided in the text. TORC1 associated with chromatin corresponds to Tor1 and Kog1 association.

**Figure 2 biomolecules-13-00642-f002:**
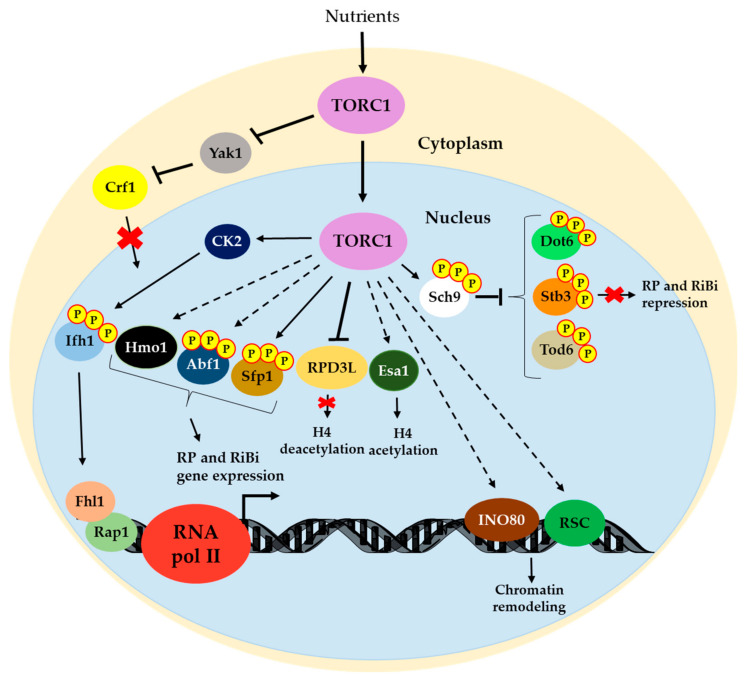
Regulatory network from TORC1 to RP and RiBi genes. TORC1 promotes RP and RiBi gene expression by maintaining the Crf1 corepressor in the cytoplasm (through Yak1 kinase) by avoiding its interaction with the Rap1/Fhl1 complex [75]. Ifh1 is phosphorylated by CK2 (downstream effector of TOR) to allow its interaction with Rap1/Fhl1 and to drive RP and RiBi gene expression [76,77]. TORC1 directly phosphorylates Sfp1 by promoting RiBi and a set of RP genes expression [81]. Abf1 could modulate some RP and RiBi genes in a TOR-dependent manner [84,85]. Hmo1 activity on RP gene expression is also modulated by TOR [51]. TORC1 also contributes to RP and RiBi gene expression by phosphorylating Sch9 that, in turn, phosphorylates repressors Dot6, Tod6 and Stb3 [89,90]. Finally, TORC1 could regulate histone modification (positively through Esa1 [95] and negatively through RPD3L [91]) and chromatin remodeling (through RSC [93] and INO80 [94]) to promote RNA pol II transcription. As Figure 1 depicts, solid lines indicate direct positive regulation. Dotted lines denote indirect positive regulation. Bars refer to negative regulation. Details for each mechanism are provided in the text.

**Figure 3 biomolecules-13-00642-f003:**
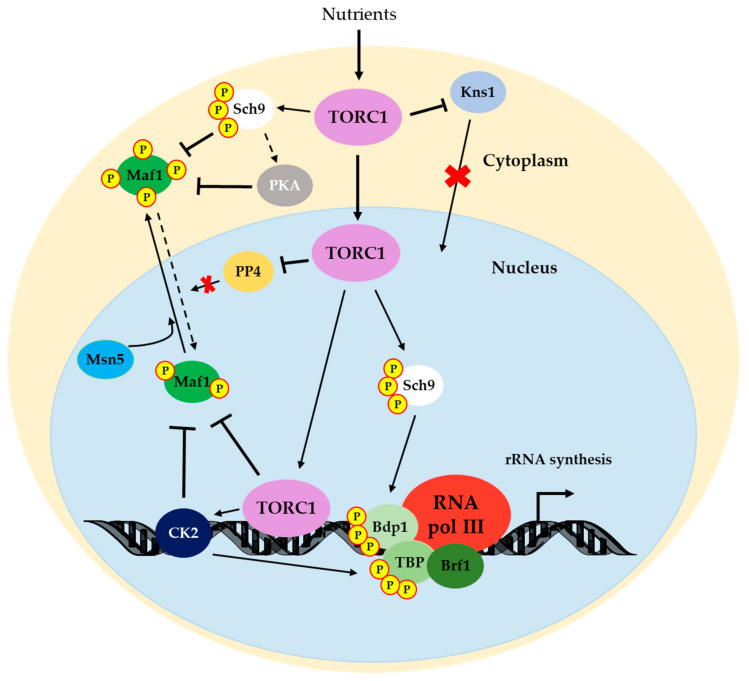
Regulatory network from TORC1 to RNA pol III. TOR promotes rRNA production for ribosome biogenesis by RNA pol III by phosphorylating Sch9 [28], which, in turn, maintains the Maf1 repressor in the cytoplasm through direct phosphorylation [40,107,108]. Crosstalk between TOR and the Ras/PKA pathway has been proposed, probably driven by Sch9 [120]. PKA also phosphorylates Maf1 [104,107]. TOR contributes to RNA pol III transcription by inhibiting PP4 activity over Maf1 [125] and by modulating Kns1 expression to avoid the phosphorylation of the Rpc53 subunit [128]. TOR also directly phosphorylates Maf1 in chromatin (as CK2), thus avoiding the Maf1-RNA pol III interaction [31,105]. Maf1 is then exported from the nucleus by Msn5 [103]. Additionally, Sch9 and CK2 phosphorylate Bdp1 [132] and TBP [124], respectively, to promote RNA pol III transcription. Note that the TFIIIC transcription factor is not represented since regulation by TOR has not yet been demonstrated. As in previous figures, solid lines indicate direct positive regulation. Dotted lines refer to indirect positive regulation. Bars denote negative regulation. Details for each mechanism are provided in the text. TORC1 associated with chromatin corresponds to Tor1 and Kog1 association.

## Data Availability

Not applicable.

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
