# Peer review of "Transcription by the Three RNA Polymerases under the Control of the TOR Signaling Pathway in Saccharomyces cerevisiae"

_biomolecules, 2023, doi:10.3390/biom13040642_

Round 1

Reviewer 1 Report

Although the manuscript by Gutierrez-Santiagio and Navarro gives very thorough overview of transcriptional regulation by Tor1 in S.cerevisiae, this article still needs some essential work on the text, to make it less floppy. English must be verified by a native speaker. I would also add a chapter that compares transcriptional regulation in yeast with those in mammals and may be other species. In this additional chapter it is not necessary to go into details, but will be important to mark the most outstanding similarities and differences in transcriptional regulation by TOR in different species. That will give more value to this review and will be more interesting to a wider spectrum or readers and not only those working in yeast.

There are many unnecessary details that makes this manuscript boring to read and eventually difficult to understand. There are so many examples! I will list few below, but authors must “clean” the text. Make your phrases simple! Avoid multiple “It has been estimated that…, it is well known that…, different studies have identified that…” etc.

Lane 60 – “It has long since been known that cells sense a wide variety of environmental cues write simply “cells respond to a wide variety of environmental cues”. By the way, it is better to say respond to a cue” than “sense a cue”.

Lane 211 – you only mention TCO89 here. In one place in the text. It is an unnecessary detail. The only important thing to say here is “Tor1 inhibition”. This will also cover “rapamycin treatment”.

Lane 381 – “northern blotting experiments have revealed” – on the review it is not important to know what exact experiment was done to demonstrate an effect. Unless, of course, if it is absolutely important, which is not the case here.

Lane 229 – “TORC1 localizes in the nucleus” – there is no data, that entire complex 1 localizes at the nucleus. Tor1 protein does. Moreover, the articles cited describes Tor1 protein localization (it is even in the title), not TOR complex 1. Please, check all through the text that nuclear Tor1 is a protein, not TORC1.

Lane 234 – please re-write avoiding underlined words: bidirectional arrows are represented in the last case due to crosstalk between that element and TOR

Lane 241 – studies state, not “states”. It is even better to use another word instead of “state” – demonstrate, show etc.

Lane 243 – Phosphorylation status regulation - delete “status”

Lane 292 – it is not clear on what your hypothesis is based on. I would delete it

Lane 296 – rDNA gene body – strange formulation, probably delete “body”

The review has many terminological inaccuracies.

Here are several examples:

1)    Lane 83-84. Authors propose to refer everything that concerns TORC1 pathway name as TOR: “TOR pathway TOR signaling TOR cascade”. I would avoid the terminology TOR cascade as it is very rarely used in the literature and almost not used in the manuscripts after 2000. But the most confusing part is to use “TOR” when it is not clear if this is referring to TOR complex 1 or to Tor1 protein. This is especially relevant to the part where functions of nuclear Tor1 discuss. To the best of my knowledge so far there is no evidence that fully functional TOR complex 1 exist in the nucleus, while Tor1 protein has been detected there.

2)    Lane 114 and all through the text authors say “RNA pol I”. It is either “RNA polymerase I” or “Pol I”. Please correct.

3)    Lane 123. “Mechanistically, rapamycin causes Rrn3-RNA Pol I dissociation”. The way how this sentence is written one could think that rapamycin can bind either to one or to another protein to trigger this dissociation.  I even went to see the original papers and found that in fact treatment with rapamycin causes “decrease of association” of this complex, which is not the same as “dissociation”

4)    Lane 190 – please find another word to replace “aforementioned”

5)    Lane 194 – “chromatinic” must be changed

6)    Lane 217 – Tor1 kinase dynamically shuttles” – delete “dynamically”

7)    Lane 223 – phosphosites were not “proposed”, they were identified.

8)    Lane 324 – “genome-wide localization profile” – change, please.

9)    The title of the chapter 6 does not reflect the content. In this chapter authors trying to discuss how Tor1 or TORC1 can regulate all three RNA Pols. Using a term “overview” gives an impression that this chapter is a summary of what was written before.

Reviewer 2 Report

This is a very comprehensive review, as far as it goes.  In this reviewers opinion, the paper would have benefitted from the inclusion of AKT and PKA  signaling.

The authors, at times, conflate yeast and mammalian rDNA transcription. For example, it has been demonstrated that rapamycin will inhibit mammalian rDNA transcription. However, it is not clear if the primary target is UBF or Rrn3. Both of these proteins are highly phosphorylated (see Cavanaugh et al (not cited bythe authors) and O’Mahony et al.), and there are reports that both are targets of TOR (Grummt and Hannan’s laboratories).  On the other hand, it has been reported that the phosphorylation of RNA polymerase I, not the phosphorylation of Rrn3, regulates rDNA transcription in yeast (see Fath, et al.).

The discussion of the TOR dependent affect on Rrn3 levels in yeast is somewhat distracting. Those authors did report that the inhibition of TOR led to the reduction in Rrn3 levels. However, they concluded that this was not sufficient to explain the decreased rate of rDNA transcription.

The manuscript suffers from numerous grammatical errors. While they do not occlude the understanding of the paper, they do detract from the manuscript.

Round 2

Reviewer 1 Report

Thank you for corrections you made in the revised version.